# Propylene Glycol Stabilizes the Linear Response of Glutamate Biosensor: Potential Implications for In-Vivo Neurochemical Monitoring

**Gaia Rocchitta [1],\*** , **Andrea Bacciu [1]**, **Paola Arrigo [1]**, **Rossana Migheli [1]**, **Gianfranco Bazzu [1]** and **Pier Andrea Serra [1,2]**

[1]  Department of Medical, Surgical and Experimental Sciences, Sassari University, Viale San Pietro 43/b, 07100 Sassari, Italy; andreabacciu90@gmail.com (A.B.); pa1989@live.it (P.A.); rmigheli@uniss.it (R.M.); gbazzu@uniss.it (G.B.); paserra@uniss.it (P.A.S.)
[2]  Mediterranean Center for Disease Control, Sassari University, Via Vienna 2, 07100 Sassari, Italy
\*  Correspondence: grocchitta@uniss.it; Tel.: +39-079-228-526

**Abstract:** L-glutamate is one the most important excitatory neurotransmitter at the central nervous system level and it is implicated in several pathologies. So, it is very important to monitor its variations, in real time in animal models' brain. The present study aimed to develop and characterize a new amperometric glutamate biosensor design that exploits the selectivity of Glutamate Oxidase (GluOx) for l-glutamate, and the capability of a small molecule as propylene glycol (PG), never used before, to influence and extend the stability and the activity of enzyme. Different designs were evaluated by modifying the main components in their concentrations to find the most suitable design. Moreover, enzyme concentrations from 100 U/mL up to 200 U/mL were verified and different PG concentrations (1%, 0.1% and 0.05%) were tested. The most suitable selected design was $Pt_c/PPD/PEI(1\%)_2/GlutOx_5/PG(0.1\%)$ and it was compared to the same already described design loading PEDGE, instead of PG, in terms of over-time performances. The PG has proved to be capable of determining an over-time stability of the glutamate biosensor in particular in terms of linear region slope (LRS) up to 21 days.

**Keywords:** L-gutamate; amperometric biosensors; propylene glycol; stability over time

## 1. Introduction

L-glutamate is one the most important excitatory neurotransmitter at Central nervous System (CNS) level, and it is involved not only in neurotransmission [1,2] but also in normal brain functioning as movement, cognitive processes, memory and plasticity [3]. So, it has become a very important compound not only in biomedical analysis but also for the implications in the field of food [4].

L-glutamate has been also demonstrated to be involved in many pathologies as epilepsy, schizophrenia, stroke and neurological disorders as Parkinson's disease [5–7], but also amyotrophic lateral sclerosis and Alzheimer's disease [8,9]. For these reasons, it has become very important to measure glutamate concentrations, and their variations, in real time in animal models.

The most widely used technique for in vivo glutamate monitoring was undoubtedly microdialysis [10–14] but electrochemical biosensors have been used for many years because of the minimal invasiveness, the rapid and real-time quantification of the studied analytes [15–17].

In addition to the previous characteristics, the biosensors, particularly in case of glutamate, require a high sensitivity in monitoring because of the glutamate low concentrations detected in the brain, assessed being in the range of 1–10 μM, and that can be dependent on the brain region of implantation [1].

One of the open challenges in biosensing is, without a doubt, the real-time glutamate monitoring in CNS in the presence of high levels of electrochemical interferents. Actually, the specificity of amperometric biosensors can be affected by interfering compounds is present in high concentration as in case for ascorbic acid (AA) at CNS level [18,19]. This is the case that occurs during the monitoring of glutamate that is found, in physiological conditions, at rather low concentrations [1].

So, because their very high temporal and spatial resolution, implantable amperometric biosensors have proved to be suitable for the implantation, from short to mid-term period, in brain tissues of animal models.

Amperometric biosensors exploit the capability of the biocomponent, usually represented by an enzyme, to specifically bind the studied analyte [17,20] and to generate some electroactive compound, converting a biological signal into a measurable electrical signal [21].

In this work, a biosensor design has been developed and characterized for the detection of glutamate, exploiting the capability of L-glutamate oxidase (GluOx) enzyme to convert L-glutamate as follows:

$$\text{L-Glutamate} + \text{H}_2\text{O} + \text{GluOx/FAD} \rightarrow \alpha\text{-ketoglutarate} + \text{NH}_3 + \text{GluOx/FADH}_2 \qquad (1)$$

$$\text{GluOx/FADH}_2 + \text{O}_2 \rightarrow \text{GluOx/FAD} + \text{H}_2\text{O}_2 \qquad (2)$$

$$\text{H}_2\text{O}_2 \rightarrow \text{O}_2 + 2\text{H}^+ + 2\text{e}^- \qquad (3)$$

The hydrogen peroxide ($\text{H}_2\text{O}_2$), produced by the enzymatic reaction, can be easily amperometrically measured at a Pt surface by applying a high fixed anodic potential of + 0.7 V vs. Ag/AgCl [22–24].

In the past, it has been highlighted that the presence of an enzymatic stabilizer as polyethileneimine (PEI) is of fundamental importance for increasing the analytical performances of glutamate biosensor [1,23,25]. In this work, the impact of the use of PEI it will be also evaluated.

Moreover, this work will be focused on the entrapment strategies needed to block all the layered biosensor components in order to ameliorate over-time performances of glutamate biosensor, both for an eventual acute and mid-term implant.

In the past, lots of approaches have been adopted to entrap enzyme molecules on the biosensor transducer surface such as covalent coupling, physical adsorption, gel entrapment or crosslinking [17,26].

About crosslinking strategy, the most commonly used for the immobilization of enzymes are polymeric films deposited on top of layered components [27–31] or glutaraldehyde crosslink [32–35]. Poly-(ethylene glycol) diglycidyl ether (PEGDE) as crosslinking agent has been proposed in the past [4] and also recently it has been used, resulting a more suitable crosslinker compared with glutaraldehyde for the immobilization of glutamate oxidase and capable to determine better biosensor sensitivities [1,35–37]. It has been demonstrated the epoxide groups present in PEGDE molecule can interact with amino groups allowing the cross-linking with the enzyme and also the covalent bonding of the enzyme layer to the surface [38].

In the past, some papers demonstrated that some compounds as amino acids, methylamines, sugars and polyols, among them propylene glycol, could act as protein stabilizers because of the hydrophobic interaction with protein molecules and because of the protection of enzyme activity [39–43]. Among all polyols, in the present paper special attention has been given to propylene glycol (PG) that is a nontoxic compound commonly used in the pharmaceutical products as a solvent or stabilizer for many compounds. In the present study it has been evaluated the role of propylene glycol as enzyme stabilizer and containment network of the layered components constituting the biosensors and its impact on kinetic and analytical parameters but also on the stability overtime of the proposed glutamate biosensor design.

Propylene glycol biosensors design has been then compared to equivalent but with PEGDE.

## 2. Materials and Methods

### 2.1. Chemicals and Reagents

All chemical compounds were bought from Sigma-Aldrich and used as supplied.

The phosphate-buffered saline (PBS, 50 mM) solution used for calibrations and electropolymerizations was made using 0.15 M NaCl, 0.05 M $NaH_2PO_4$ and 0.04 M NaOH, and adjusted to pH 7.4.

Sodium glutamate (Glut, 1 M and 10 mM), PG, PEGDE (0.1%), polyethyleneimmine (PEI, 1%) solutions were diluted in double-distilled water, while o-phenylenediamine (OPD) was prepared in fresh PBS. Stock solutions of ascorbic acid (AA, 100 mM) were prepared in HCl 0.01 N and stored at −20 °C.

Glutamate Oxidase (GluOx, Streptomyces sp., EC 1.4.3.11, 200 U/mL) was from Yamasa Corp.

Teflon insulated Platinum/Iridium wire (Pt, 90:10, Ø = 125μm) was bought from Advent Research Materials (Eynsham, UK).

### 2.2. Instrumentation and Software

For all electrochemical experiments, a conventional three-electrode cell was used which consisted of a beaker holding 20 mL of fresh PBS, four glutamate biosensors as working electrodes, an Ag/AgCl (3M) electrode (Bioanalytical Systems, Inc. West Lafayette, IN, USA) and a large surface stainless steel needle as auxiliary electrode. A four-channel potentiostat (eDAQ Quadstat, e-Corder 410, eDAQ Europe, Poland) was utilized for all electrochemical procedures. All potentials used in this work refer to the aforementioned reference electrode.

### 2.3. Biosensor Construction and Characterisation

As shown in Figure 1, different biosensor designs were developed, all of them based on a cylindrical geometry (1 mm length and 125 μm diameter).

Biosensors were manufactured as previously described [1,23,24] and realizing different strategies differentiating biosensor designs, varying components concentrations and processes, to obtain the most performing.

Briefly, a 3 cm portion of Pt wire was cut and from one edge, 3 mm of insulation was eliminated in order to allow bare metal to be welded to its support. From the other edge of the wire, a 1 mm of bare metal was exposed for modifications.

First, on Day 0, PPD electropolymerization was carried out by immersing the Pt wires in an OPD 300 mM solution freshly prepared in PBS, and by applying a positive potential of + 0.7 V vs. Ag/AgCl (3M) for 30 min.

After having rinsed electrodes in pure water, they were immersed twice in a PEI 1% solution, waiting 5 minutes between each immersion. Then, electrodes were dipped 5 times in the GlutOx solution, at different concentrations (dependently on the chosen design), allowing 5 min from one dip and another to allow the drying of each enzyme layer. On top, a layer of PG, at different concentrations, (dependently on the chosen design), or PEGDE, was deposited by a single dip, allowing the layer to dry for 30 min. At the end of procedures, biosensors were immersed in fresh PBS and a positive potential of + 0.7 V vs. Ag/AgCl was applied to allow an overnight stabilization of the currents.

On Day 1 a full glutamate calibration, ranging between 0 and 100 mM, was carried out in fresh PBS by the addition of known volumes of the Glut stock solutions (1 M and 10 mM). On same day an AA calibration ranging from 0 to 1000 μM was performed to assess the design capability to block currents from interfering compounds. Each biosensor, in all designs, was subjected to the same calibrations' protocol that was repeated on Day 7, and for some of them appropriately selected, also at Day 10, 15 and 21 to assess biosensor ageing.

Different designs have been manufactured (Figure 1): the basic design (D1) involved the deposition of all the components previously described except the PEI. A second design (D2) was

made by the deposition of all components and above all PG (0.1%). Starting from D2 design, some variations on GluOx and PG concentrations were performed and characterized. A third design (D3) was built by depositing above all layered components PEGDE (0.1%). Biosensor performances were evaluated from calibration data in terms of enzymatic kinetic ($V_{MAX}$ and $K_M$) and analytical performances (linear region slope—LRS, AA blocking).

Each biosensor design, after every calibration, was rinsed in double-distilled water and then stored at + 4 °C when not in use.

All designs were tested for currents due to the main interfering compounds as DA, DOPAC, UA, 5-HIAA. In all cases, the above-mentioned compounds did not determine any interfering signal on biosensor glutamate monitoring, as previously showed [44].

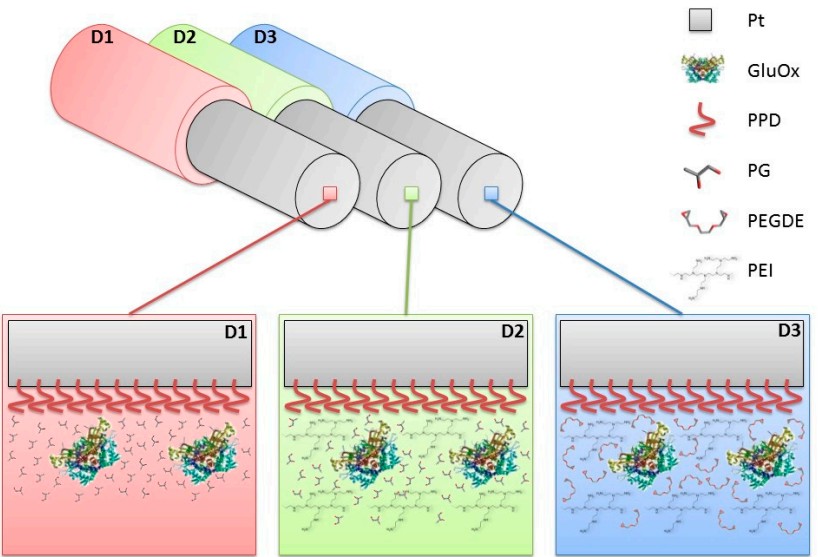

**Figure 1.** Schematic representation of the main designs of glutamate biosensors characterized in this study. **D1**: $Pt_c$/PPD/GluOx5/PG (0.1%); **D2**: Ptc/PPD/PEI(1%)$_2$/GluOx$_5$/PG(0.1%); **D3**: $Pt_c$/PPD/PEI(1%)/GluOx$_5$/PEGDE(0.1%). $Pt_c$: Pt cylinder 1 mm long, 125 μm diameter; GluOx: L-glutamate oxidase; PPD: poly-ortho-phenylenediamine; PEI: polyethyleneimine; PG: propylene glycol; PEGDE: Poly(ethylene glycol) diglycidyl ether. The subscript number represents the number of dip−evaporation deposition steps and in brackets the concentration of the component.

## 2.4. Statistical Analysis

After calibrations, biosensor currents were plotted versus Glut and AA concentrations. First, for glutamate data a nonlinear fitting with Michaelis–Menten equation was performed on the entire concentration range (0–100 mM) to evaluate $V_{MAX}$ and apparent $K_M$. Linear regressions (slope) were calculated at low concentrations (0–0.6 mM). Recorded currents were given in nanoamperes (nA) and expressed as baseline-subtracted values ± standard error of the mean.

AA shielding was evaluated by taking in account the current recorded at 500 μM and ΔI value, which stands for the difference between the current resulting from the injection of 1000 μM and 500 μM of AA in the electrochemical cell, as discussed previously [45].

Statistical significance (P values) between groups was assessed using unpaired t-tests by GraphPad Prism 5.02 v software.

The limit of detection (LOD) was calculated from the standard deviation (σ) of the response and the LRS of the calibration curve as follows [46]:

$$LOD = 3.3 \, \sigma / LRS$$

## 3. Results and Discussion

### 3.1. In Vitro Performances of Glutamate Biosensors

As shown in Figure 2 and Table 1, three different glutamate biosensor designs were compared. We opted to compare, as shown in the bar chart, the different biosensor designs (n = 4 for each group) in terms of $V_{MAX}$ (Panel A), $K_M$ (Panel B), LRS (Panel C), while other parameters as LOD, $I_{lim}$ and $\Delta I$ were showed in Table 1.

The D1 design produced the worst performances, when compared with D2 and D3 designs. In fact, D1 showed the lowest $V_{MAX}$ and LRS and the highest $K_M$, proving to have a low number of active molecules on the surface and showing an apparently low affinity of the glutamate for the enzyme. What is more, those parameters got worse on Day 7.

Conversely, D2 and D3 designs displayed interesting features in terms of $V_{MAX}$ and LRS, that resulted both higher than those measured in D1. Though, while D3 did not show any significant variation for both parameters, between Day 1 and Day 7, mainly even because $K_M$ did not undergo any significant changes, D2 suffered a significant increase in $V_{MAX}$ while showed a significant decrease in LRS, due principally to a significant increase in $K_M$. From these results, it is clear that the presence of PEI, that is not present in D1 design, is fundamental in order to increase the enzymatic activity and make the biosensor more performing, as previously demonstrated [23,25]. In fact, the polycation PEI has been demonstrated to be able to stabilize different enzymes because of the reduction of the repulsion between anionic charges on GluOx molecules, determining a better biosensor efficiency [23,25].

As can be seen from the data in Table 1, all the designs proved to have good shielding proprieties against AA at Day 1, which worsened at Day 7, to a greater extent in D1. So, an improvement in shielding towards interfering molecules should be necessary.

As can be seen from Table 1, about LOD values, no substantial differences among the studied designs were highlighted, proving that such biosensor designs would be suitable for in-vivo applications where physiological glutamate concentrations have been assessed to be in the low micromolar range [1].

All data about the observations made about this experiment, as Michaelis–Menten plots and summary table containing all the enzymological and analytical parameters (Figure S1 and Table S1), are reported in the Supplementary Materials section.

From these data, it is possible to deduce an interesting observation on the molecule used as a retention net of the biosensor layers.

The crosslinker PEGDE has been largely used in biosensor fabrications and in particular for glutamate biosensors [1,35–37]. Our results, in the considered range of time, confirm the capability of PEGDE to avoid the loss of substrate specificity that can occur during enzyme immobilization. It has been demonstrated that this phenomenon is due to its mild action towards the enzyme molecules, as a result of its epoxy moiety on enzyme amino groups, thus saving enzyme conformation [37]. Surprisingly, a small molecule as PG is, which is not a polymer and that was never used before in biosensor manufacture, has shown to own characteristics completely overlapping with those of PEGDE. Actually, also PG was able to avoid the loss of substrate specificity, in fact, $V_{MAX}$ increased between Day 1 and Day 7.

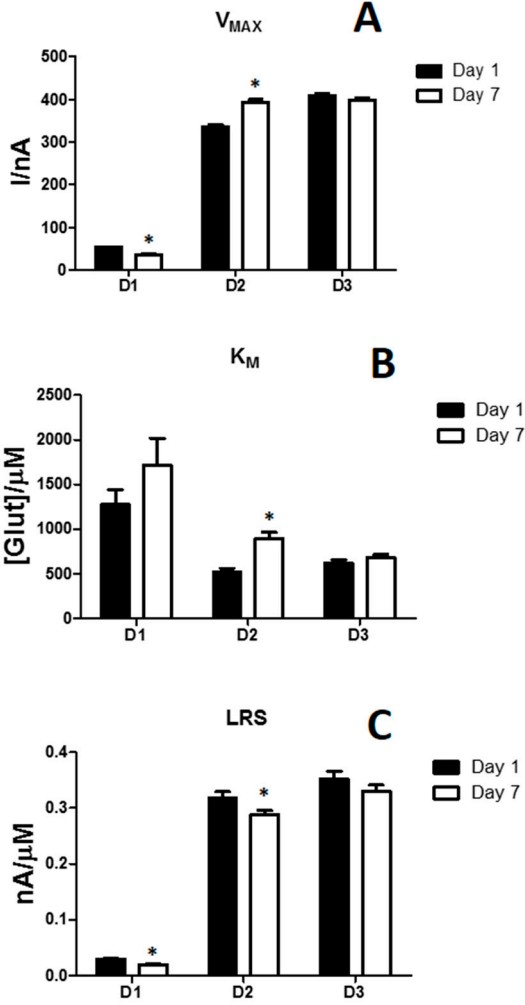

**Figure 2.** Bar chart describing the variation of $V_{MAX}$ (Panel A), $K_M$ (Panel B) and LRS (Panel C) on **D1**, **D2** and **D3** biosensor designs at Day 1 (black bars) and Day 7 (withe bars). Values are expressed as mean $\pm$ SEM. * $p < 0.05$ vs. Day 1. **D1**: $Pt_c$/PPD/GluOx5/PG (0.1%); **D2**: $Ptc$/PPD/PEI(1%)$_2$/GluOx$_5$/PG (0.1%); **D3**: $Pt_c$/PPD/PEI(1%)$_2$/GluOx$_5$/PEGDE (0.1%). $Pt_c$: Pt cylinder 1 mm long, 125 μm diameter; GluOx: L-glutamate oxidase; PPD: poly-ortho-phenylenediamine; PEI: polyethyleneimine; PG: propylene glycol; PEGDE: Poly(ethylene glycol) diglycidyl ether. The subscript number represents the number of dip−evaporation deposition steps and in brackets the concentration of the component.

**Table 1.** In vitro characterization of three different biosensor designs at Day 1 and day 7 in terms of Michaelis–Menten kinetic parameters ($V_{MAX}$ and $K_M$) and analytical parameters (Linear Region Slope –LRS, and LOD) for each design. **D1**: $Pt_c$/PPD/GluOx5/PG (0.1%); **D2**: Ptc/PPD/PEI(1%)$_2$/GluOx$_5$/PG (0.1%); **D3**: $Pt_c$/PPD/PEI(1%)$_2$/GluOx$_5$/PEGDE(0.1%). $Pt_c$: Pt cylinder 1 mm long, 125 μm diameter; GluOx: L-glutamate oxidase; PPD: poly-ortho-phenylenediamine; PEI: polyethyleneimine; PG: propylene glycol; PEGDE: Poly(ethylene glycol) diglycidyl ether. The subscript number represents the number of dip−evaporation deposition steps and in brackets the concentration of the component.

| Parameters | D1 | D2 | D3 |
|---|---|---|---|
| | DAY 1 | | |
| $V_{MAX}$ (nA) | 52.71 ± 1.70 | 335.91 ± 5.03 | 409.73 ± 5.58 |
| $K_M$ [μM] | 1275.02 ± 163.81 | 524.53 ± 37.62 | 620.23 ± 37.65 |
| LRS (nA/μM) | 0.030 ± 0.001 | 0.318 ± 0.011 | 0.352 ± 0.014 |
| $I_{lim}$ (nA) | 1.984 ± 0,056 | 1.450 ± 0.124 | 0.867 ± 0.074 |
| ΔI | −0.027 | 0.482 | 0.388 |
| LOD [μM] | 0.265 | 0.252 | 0.267 |
| | DAY 7 | | |
| $V_{MAX}$ (nA) | 36.23 ± 1.67 | 392.16 ± 7.73 | 399.5 ± 4.88 |
| $K_M$ [μM] | 1717.12 ± 299.12 | 892.84 ± 73.55 | 680.34 ± 34.85 |
| LRS (nA/μM) | 0.020 ± 0.001 | 0.288 ± 0.008 | 0.330 ± 0.011 |
| $I_{lim}$ (nA) | 26.072 ± 12.178 | 2.153 ± 0.430 | 4.127 ± 1.103 |
| ΔI | 15.103 | 1.061 | 1.848 |
| LOD [μM] | 0.270 | 0.221 | 0.257 |

*3.2. Effects of the Difference of Enzyme Concentration in Glutamate Biosensor Parameters*

In this study, different concentrations of enzyme were used. As displayed in Figure 3, $V_{MAX}$ (Panel A), $K_M$ (Panel B) and LRS (Panel C) values were showed, at Day 1 (black bars) and at Day 7 (white bars). From the chart, it can be seen how $V_{MAX}$ was dependent on the enzyme concentration. In fact, 200 U/mL design showed a $V_{MAX}$ equal to 335.91 ± 5.03 nA, while 150 U/mL and 100 U/mL designs displayed 232.81 ± 5.65 nA and 98.19 ± 7.67 nA respectively, highlighting how the number of active molecules on the surface of the biosensor strongly depends on the enzymatic load. As expected, also $K_M$ was influenced by the loaded enzyme on the surface. Surprisingly, at Day 1, the 100 U/mL design, produced the highest $K_M$, equal to 1088.03 ± 348.90 μM, while 150 and 200 U/mL designs showed the same $K_M$ values, of about 500 μM. But, though 150 and 100 U/mL did not suffer any significant variation between Day 1 and Day 7, the 200 U/mL underwent a significant increase. $V_{MAX}$ and $K_M$ variations were reflected on the respective LRS, also dependently on the enzymatic load. In fact, the design loading 100 U/mL presented the lowest LRS, if compared with 150 and 200 U/mL designs. The latter design proved to be the most performing because, at Day 1, it did show the highest LRS, equal to 0.318 ± 0.011 nA/μM, which underwent a small, although significant, decrease at Day 7.

From these results, it is clear that enzymatic loading influences the biosensor response. As expected, at Day 1, the decreasing in enzyme loading determined a variation in kinetic and analytic parameters. In fact, as previously demonstrated [28,29,47], $V_{MAX}$ reflected the amount of active enzyme molecules on the transducer surface and this parameter varied in an almost linearly manner dependently on the enzyme amount on the biosensor (data not shown, $R^2$ = 0.951). Over time, it was observed that the presence of PG determined slight variation about LRS, while produced an interesting increase in $V_{MAX}$. Conversely, $K_M$ showed different, no-predictable, trends, in fact if in

200 U/mL design it decreased, in 150 U/mL design it increased and in 100 U/mL design it remained almost the same.

All complete data set about this experiment, as Michaelis–Menten plots and summary table containing all the enzymological and analytical parameters (Figure S2 and Table S1), are reported in the Supplementary Materials section.

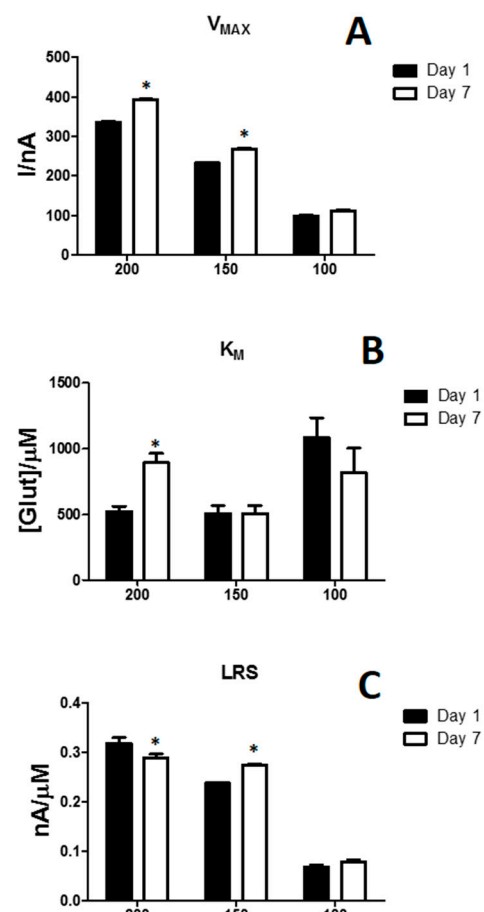

**Figure 3.** Bar chart showing the modifications of $V_{MAX}$ (Panel A), $K_M$ (Panel B) and LRS (Panel C) on biosensor designs loading different enzyme concentrations (200 U/mL, 150 U/mL and 100 U/mL) at Day 1 (black bars) and Day 7 (withe bars). Values are expressed as mean $\pm$ SEM. * $p < 0.05$ vs. Day 1.

### 3.3. Effects of the Variation of Propylene Glycol in Glutamate Biosensor Performances

As shown in Figure 4, the modification of propylene glycol concentration is highlighted in terms of $V_{MAX}$ (Panel A), $K_M$ (Panel B) and LRS (Panel C) variations determined variations in biosensor performances, from Day 1 (black bars) to Day 7 (white bars).

As shown in panel A, $V_{MAX}$s and their variations were highly influenced by the PG concentration. In fact, the design loading 0.1% of PG, at Day 1, showed the highest $V_{MAX}$ value, equal to 335.91 $\pm$ 5.03 nA. Moreover, surprisingly, in the latter design $V_{MAX}$ suffered a significant increase, between Day 1 and Day 7, unlike what happened in the other two designs where the $V_{MAX}$ decreased significantly at Day 7. In some way, to be elucidated, PG (0.1%) was able, at least formally, to increase the number of active molecules on the biosensor surface.

Even $K_M$s were influenced by PG concentrations. At Day 1, all $K_M$s were similar and around 500 $\mu$M, and all suffered a significant increase at Day 7, if compared with Day 1. Unfortunately, in the 0.05% PG design, the increase was quantitatively considerably higher, denoting an important loss of the apparent affinity of the glutamate for the enzyme.

About LRS, the 0.1% PG design showed the highest value at Day 1, that was equal to $0.318 \pm 0.011$ nA/$\mu$M, which suffered the smallest, although significant, decrease at Day 7 reaching $0.288 \pm 0.008$ nA/$\mu$M.

From these results it was possible to deduce that both halving and decupling PG concentrations did not produce the best conditions for biosensor monitoring. In fact, a global worsening for all parameters was observed. So, the concentration of 0.1% was chosen to study the behavior of the biosensor over a period of 21 days.

All complete data set about this experiment, as Michaelis–Menten plots and summary table containing all the enzymological and analytical parameters (Figure S3 and Table S2), are reported in the Supplementary Materials section.

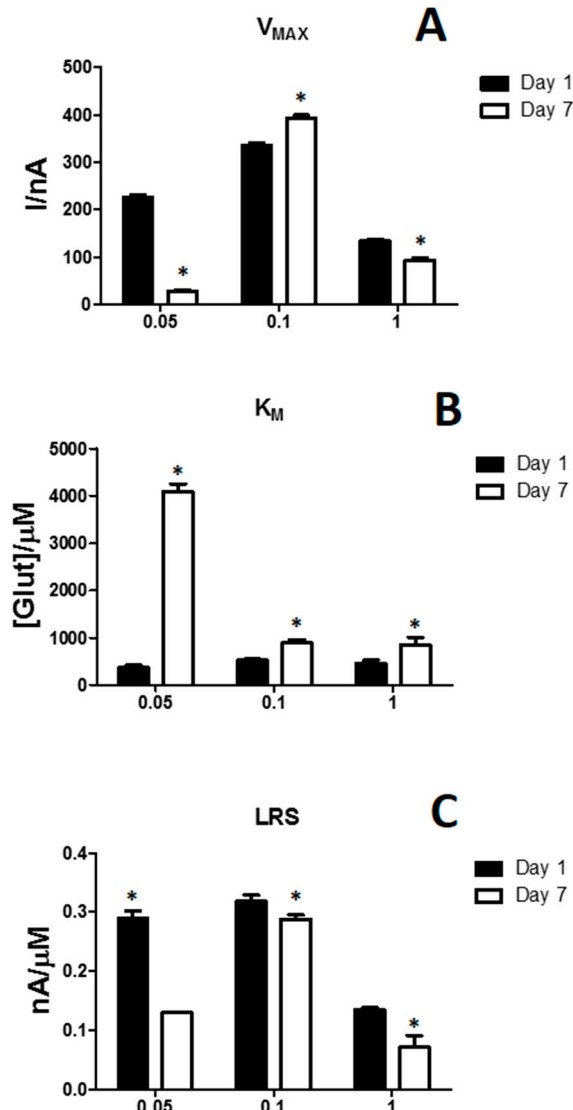

**Figure 4.** Bar chart displaying the modifications of $V_{MAX}$ (Panel A), $K_M$ (Panel B) and LRS (Panel C) on biosensor designs loading different PG concentrations (0.05%, 0.1% and 1%) at Day 1 (black bars) and Day 7 (withe bars). Values are expressed as mean $\pm$ SEM. * $p < 0.05$ vs. Day 1.

### 3.4. In Vitro Assessment of Continuous 48 h Glutamate Measurement in PBS

To assess biosensor stability for an eventual in-vivo use, a 48-h glutamate measurement was done, in order to simulate a continuous monitoring as in implant occurs. Because of the obtained results,

we decide to subject the PG (0.1%) design to this protocol. Results were then compared with PEGDE (0.1%) design, once subjected to the same protocol.

At Day 1, after baseline stabilization in fresh PBS, 2 $\mu$M injections were done every two hours, covering a range between 0–10 $\mu$M in order to simulate, in same way, glutamate spiking. PG (0.1%) design showed a current of 0.296 $\pm$ 0.009 nA per $\mu$M of injected glutamate, while the current for PEGDE (0.1%) design was 0.333 $\pm$ 0.013 nA per $\mu$M of injected glutamate. At Day 2, the same protocol was realized, yielding a current of 0.320 $\pm$ 0.011 nA/$\mu$M and 0.372 $\pm$ 0.014 nA/$\mu$M for PG (0.1%) and PEGDE (0.1%) design respectively. Both designs showed an interesting stability over the 48-h selected period, proven to be rather suitable for monitoring glutamate during a short-term implantation. Moreover, the biosensors respected the upward trend, although not significant, previously highlighted between Day 1 and Day 7.

*3.5. Effects of PG or PEGDE on Glutamate Biosensor Ageing*

Following the interesting results, the ageing of two selected designs was evaluated. The chosen designs were the one loading PG (0.1%) and the other loading PEGDE (0.1%). As shown in Figure 5, a period of 21 days was evaluated, when for each design a full glutamate calibration was performed. As highlighted in Panel A, $V_{MAX}$ variations of PG design (white columns) underwent a general increase up to Day 15 reaching 368.30 $\pm$ 8.11 nA. A non-significant decrease was obtained at Day 21 when $V_{MAX}$ reached 326.6 $\pm$ 11.14 nA.

Conversely, although the PEGDE design (black columns), at first, showed higher values of $V_{MAX}$, if compared with PG design, then, after a substantial stability between Day 1 and Dy 7, when $V_{MAX}$ was about 400 nA, it showed a general significant decrease up to Day 21 reaching 310.12 $\pm$ 5.32 nA.

In Panel B, $K_M$ over-time fluctuations are displayed. In both designs an overall increase was observed, but PG design displayed higher $K_M$s, if compared with PEGDE design, up to Day 15, when also it suffered of a significant increase if compared with Day 1. Surprisingly, at Day 21 a decrease in $K_M$ occurred in the PG design, unlike the PEGDE design, which suffered a significative increase, if compare with Day 1.

In Panel C over-time variation of LRS are shown. In PG design, despite a slight decrease between Day 1 and Day 7, when the parameter was about 0.300 nA/$\mu$M, LRS surprisingly remained substantially constant up to Day 15, in particular a little less than then0.300 nA/$\mu$M, at Day 21 underwent a slight, but not significant, decrease reaching 0.253 $\pm$ 0.004 nA/$\mu$M.

In PEGDE design, a generalized decrease in LRS values, in the considered range of time, occurred. The highest sensitivity was reached at Day 1 when LRS was 0.369 $\pm$ 0.019 nA/$\mu$M while at Day 21 LRS was 0.214 $\pm$ 0.003 nA/$\mu$M. If compared with Day 1, LRS figures from day 10 up to day 21 resulted significantly lower, if compared with Day 1.

In panel D, LOD variations in a range of time of 21 days are displayed. As can be seen from the bar chart, no significant variations occurred for PG design, while a significant variation, if compared with Day 1, occurred in PEGDE group at Day 15 and Day 21, denoting a worsening in the capability of this design to retain its response to low glutamate concentrations.

The use of PEDGE in glutamate biosensors have been widely described, its capability to stabilize enzyme molecules as well [1,4,35–37]. It has been shown that the cross-linking between PEGDE molecules and the enzyme is favored by the presence of the epoxide groups, whom can interact with amino groups allowing the covalent bonding of the enzyme layer to the surface [38]. In the present study, we have demonstrated that glutamate biosensors loading PEDGE (0.1%) show very good analytical performances over the period of 21 days as previously described [1], as well as the design loading PG (0.1%). Although PEGDE showed quite interesting values in terms of $V_{MAX}$, $K_M$ and LRS, PG design showed a very fascinating behavior over time. In fact, $V_{MAX}$ values had a surprising upward trend up to Day 15, indicating that the number of active molecules on the transducer surface was formally increased during the days. Conversely, in the PEGDE design, after a certain stability

between Day 1 and Day 7, $V_{MAX}$ showed a tendency to decrease over time, denoting a loss in the number of active molecules.

On the other hand, $K_M$ behavior was, more or less, comparable between both designs. In fact, in both cases a general increase over time was achieved, as expected.

The changes in these parameters were reflected in the LRS, which proved to be very interesting.

Nevertheless, although PEGDE design revealed higher values of LRS, showing a relatively-better efficiency in monitoring glutamate, but with a tendency to decrease over time, the PG design resulted in a sustained stability over a period of 15 days. In both cases, after 21 days, efficiency began to change significantly.

Thus, in the present study it has been highlighted that the presence of PG it turned out to be very important for glutamate biosensor in-vitro stability over time. In literature, it has been widely demonstrated the capability of molecules like propylene glycol, what polyols are, to stabilize proteins and further their long-term stability: in fact, the presence of polyols reduces the unfavorable interaction present between the solution and the proteins [48]. It has been also demonstrated that stabilizing osmolytes, as propylene glycol is, are able to move the equilibrium of proteins from denaturation to the native conformation, leading to the stabilization of the protein itself [49]. Thus, the presence of the osmolytes can change the amount of water near the protein during the reactions, determining changes also in its stability [50]. In the present study, it seems that PG has been able to interact with the proteins molecules by determining an improvement of their conformational state and producing an increase of the active molecules on the transducer surface and thus favoring the stability over time.

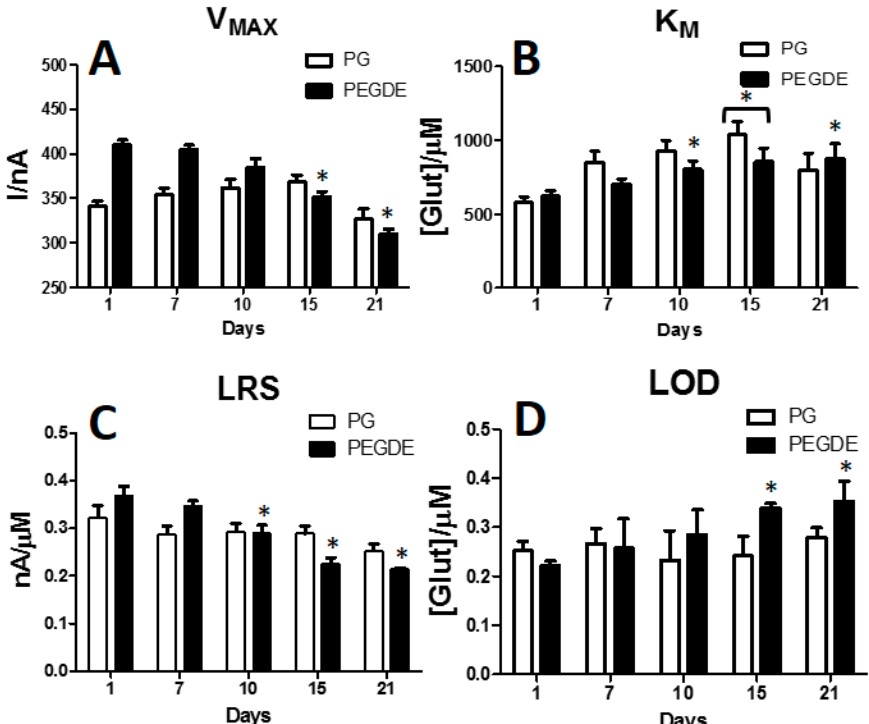

**Figure 5.** Bar chart describing the variations of $V_{MAX}$ (Panel A), $K_M$ (Panel B), LRS (Panel C) and LOD (Panel D) of the two selected design $Pt_c/PPD/PEI(1\%)_2/GlutOx_5/PG(0.1\%)$ (PG (0.1%)-white bars) and $Pt_c/PPD/PEI(1\%)_2/GlutOx_5/PEGDE(0.1\%)$ (PEGDE (0.1%) -black bars) in the selected range of 21 days. Values are expressed as mean ± SEM. * $p < 0.05$ vs. Day 1.

## 4. Conclusions

In the present study, the role of PG has been evaluated. From results it is possible to affirm that PG helps avoiding the loss of active molecules on transducer surface, on the contrary, apparently it is

able to increase their number, but also to preserve substrate specificity, making the proposed biosensor design suitable for short- and mid-term implants in animal models.

While polymeric compounds are commonly used in order to entrap all the layered components of the biosensor, as stated before, in the present paper the use of a small intercalating molecule as propylene glycol has been used. PG has never been used before for biosensor manufacture, moreover, no citations have been found about polymerization of PG in the experimental conditions used in this investigation. So, the exerted capability of PG of retaining enzyme molecules, as well as their activity, must be probably related to the intermolecular bonds established with the enzyme molecules.

The use of PG as containment net needs to be further investigate, but its capability of preserving LRS, hence the sensitivity of the biosensor over time, is the most important feature in view of a possible prolonged implant for several days, being able to guarantee the most reliable glutamate measurement possible.

At the moment, some experiments are underway to evaluate the role of PG on other biosensor, as glucose and lactate biosensors.

**Supplementary Materials:** The following are available online at http://www.mdpi.com/2227-9040/6/4/58/s1, Figure S1: Different biosensor designs Michaelis-Menten plots, Figure S2: Michaelis-Menten plots of different biosensor designs loading different GluOx concentrations, Table S1: Kinetic and analytical parameters of Figure S2, Figure S3: Michaelis-Menten plots of different biosensor designs loading different PG concentrations, Table S2: Kinetic and analytical parameters of Figure S3.

**Author Contributions:** G.R. conceived the project, defined experimental protocol and compiled the draft; A.B. and P.A. carried out experiments, G.B. and R.M. drew plots and performed statistical analysis; P.A.S. contributed to define experimental protocol and to compile the draft.

**Acknowledgments:** We thank Robert D. O'Neill for inspiration regarding the glutamate biosensor. We thank him for his teachings, but, above all, for helpful and substantive discussions.

**Conflicts of Interest:** The authors declare no conflict of interest.

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
