# Peer review of "Propylene Glycol Stabilizes the Linear Response of Glutamate Biosensor: Potential Implications for In-Vivo Neurochemical Monitoring"

_chemosensors, doi:10.3390/chemosensors6040058_

Reviewer 1 Report

The manuscript reported the development and the characterization of a new glutamate biosensor with improved Glutamate Oxidase stability and activity influenced by small compound like Propylene Glycol.The presentation of the data are well organized and their discussion is convincing. The manuscript is acceptable for publication after minor revision: Following there are some comments:

1.    The authors report the comparison of three different glutamate biosensors in terms of kinetic and analytical parameters at Day1 and Day 7. The limit of detections (LOD) are absents. Discuss about this and add the value in table 1 and in Result and discuss section. 

2.    Add in the Figure 5 a bar chart describing the variations of the LODs in the selected range of 21 days.

3.    In the line 237, please correct value using the same significant figures.

4.     In the line 97 modify the NaH2PO4 in NaH2PO4and in the line 101 HCL in HCl

Author Response

We would like to thank the referee for the observation on the manuscript. The critical issues highlighted were resolved as follows:

1.  The authors report the comparison of three different glutamate biosensors in terms of kinetic and analytical parameters at Day 1 and Day 7. The limit of detections (LOD) are absents. Discuss about this and add the value in table 1 and in Result and discuss section.

As requested from the referee, LODs have been added in Table 1 and in results and discussion section. Moreover, the part concerning LOD calculation has been added in “Statistical Analysis” paragraph with the related reference citation, thus the change in listing references has been done and highligthed along the text.

In the results and discussion section, in paragraph “3.1 In Vitro performances of glutamate biosensors” it has been added the following sentence: “As can be seen from table 1, about LOD values, no substantial differences among the different designs were highlighted, demonstrating that such biosensor designs would be suitable for in-vivo applications where physiological glutamate concentrations have been assessed to be in the low micromolar range [1].”

 2.Add in the Figure 5 a bar chart describing the variations of the LODs in the selected range of 21 days.

As requested from the referee, LODs have been added in Figure 5 and discussed. In the results and discussion section, in paragraph “3.3 Effects of PG or PEGDE on glutamate biosensor ageing” the following text was added “In panel D, LOD variations in a range of time of 21 days are displayed. As can be seen from the bar chart, no significant variations occurred for PG design, while a significant variation, if compared with Day 1, occurred in PEGDE group at Day 15 and Day 21, denoting a worsening in the capability of this design to retain its response to low glutamate concentrations.”

Moreover, the part concerning LOD calculation has been added in “Statistical Analisys” paragraph with the related reference citation, thus the change in listing references has been done and highligthed along the text

3.   In the line 237, please correct value using the same significant figures:

the number 11.31 ± 3.634 nA has been changed in 11.315 ± 3.634 nA

4. In the line 97 modify the NaH2PO4 in NaH2PO4 and in the line 101 HCL in HCl:

In line 97 and in line101 the requested changes have been made.

Reviewer 2 Report

The work of Rocchitta et al. reports on the evaluation of the effect of Propylene glycol (PG) on the performance of the glutamate biosensors.

The work is interesting and experimentally solid, but some significant modifications are, in the view of the reviewer, needed to make it appealing for publication.

One of the core objectives of the work is to evaluate the performance of different sensor architecture as the function of the storing time. Unfortunately, the authors failed to describe crucial experimental points as:

1)     How the sensors were stored?

2)     Were the same sensors used at day 1 day 7 and following days or fresh sensors (prepared at day 0 and stored) were used in the studies?

The authors claim that the use of PG in sensor architecture is beneficial for their future use in implanted system. In the manuscript the authors only show improvement in storage time of sensors. To demonstrate the benefit in implantable systems the authors should present results of the performance of their best PG-containing sensor architecture on a long-time measurement experiment simulating operation in implanting (e.g. measuring glutamate addition at different time while having the sensor running in buffer for at least 24 hours) and compare them with a PG free sensor architecture.

The result and discussion section need to be revised.

This section is often reduced to a list of numbers (all relevant) but what is missing an attempt to understand/explain the obtained results.

The current structure makes the reading not appealing and does not provide, in most of the case, to the reader (in the view of the reviewer) the information/consideration that will make the findings reported applicable to other works.

The authors should try to rethink the way results are reported and described using, for example similar graphical presentation as in Figure 5 and list the different obtained value in a table (supporting info?).

Finally, in conclusions and abstract the authors should highlight better the novelty of the proposed work.

Author Response

The work of Rocchitta et al. reports on the evaluation of the effect of Propylene glycol (P G) on the performance of the glutamate biosensors.

The work is interesting and experimentally solid, but some significant modifications are, in the view of the reviewer, needed to make it appealing for publication.

One of the core objectives of the work is to evaluate the performance of different sensor architecture as the function of the storing time. Unfortunately, the authors failed to describe crucial experimental points as:

1) How the sensors were stored?

We would like to thank the reviewer for the observation he/she made. Actually, we forgot to mention that the “Each biosensor, after every calibration, was rinsed in double distilled water and then stored at +4°C when not in use. This part of the protocol has been added in Materials and Method section in the paragraph “2.2 Biosensor construction and characterisation

2)  Were the same sensors used at Day 1 Day 7 and following days or fresh sensors (prepared at day 0 and stored) were used in the studies?

We would like to thank the reviewer for this observation. Actually, we were not sufficiently clear in explaining the protocol.

Obviously, the biosensors were always the same when the calibrations were effectuated at Day 1, Day 7 and following. This point has been better elucidated in Materials and Method section in the paragraph “2.2 Biosensor construction and characterisation” as follows:

For all designs, the same calibrations’ protocol was repeated on Day 7, and for some of them appropriately selected, also at Day 10, 15 and 21 to assess biosensor ageing.” has been changed in “Each biosensor, in all designs, was subjected to the same calibrations’ protocol that was repeated on Day 7, and for some of them appropriately selected, also at Day 10, 15 and 21 to assess biosensor ageing.”

The authors claim that the use of PG in sensor architecture is beneficial for their future use in implanted system. In the manuscript the authors only show improvement in storage time of sensors. To demonstrate the benefit in implantable systems the authors should present results of the performance of their best PG-containing sensor architecture on a long-time measurement experiment simulating operation in implanting (e.g. measuring glutamate addition at different time while having the sensor running in buffer for at least 24 hours) and compare them with a PG free sensor architecture.

We would like to thank the referee for this remark about the response of the biosensors during implantation.

To try to meet the referee's request, a short-term 48 h experiment was performed in PBS.

As it would not have been possible to build a PG free biosensor, without altering substantially its response, because the design would be completely different. Thus, the authors decided to use the biosensors loading PG (0.1%) and PEGDE (0.1%) for the requested experiment. Results and discussions obtained have been discussed in the Results and discussions section, in the paragraph” 3.5 In vitro assessment of continuous 48 h glutamate measurement in PBS”

The result and discussion section need to be revised.

This section is often reduced to a list of numbers (all relevant) but what is missing an attempt to understand/explain the obtained results.

The current structure makes the reading not appealing and does not provide, in most of the case, to the reader (in the view of the reviewer) the information/consideration that will make the findings reported applicable to other works.

The authors should try to rethink the way results are reported and described using, for example similar graphical presentation as in Figure 5 and list the different obtained value in a table (supporting info?).

We would like to thank the referee for this valuable observation. In agreement with his/her suggestions, all the figures, and relative legends, were changed from Michaelis-Menten kinetics plots into bar charts, with the aim of promoting data comprehension. A statistical analysis was also added. The Michaelis-Menten kinetics plots with the relative summary tables, reporting VMAX, KM, LRS, Ilim, ΔI were inserted in the Supporting Material section.

All the results and discussion in the following paragraphs, “3.1 In Vitro performances of glutamate biosensors”, “3.2 Effects of the difference of enzyme concentration in glutamate biosensor parameters” and “3.4 Effects of PG or PEGDE on glutamate biosensor ageing” were changed with the aim of promoting experiments’ comprehension.

Also, the Conclusion section was revised trying to better elucidate the obtained results and to explain the further applicability of the proposed design for possible experiments in animal models.

Finally, in conclusions and abstract the authors should highlight better the novelty of the proposed work.

Welcoming the last referee's request, a variation in the abstract was made about the novelty of the paper. Actually, the part “Until now, different polymeric compounds have been used to contain all the components constituting the biosensor, while in this study a small intercalating molecule as PG, never used before, was used. Surprisingly, PG determined an over-time stability of the glutamate biosensor in particular in terms of linear region slope (LRS) up to 21 days.” was added.

In the Conclusion section it has been added this statement “While polymeric compounds are commonly used in order to entrap all the layered components of the biosensor, as stated before, in the present paper the use of a small intercalating molecule as propylene glycol has been used. PG has never been used before for biosensor manufacture, moreover, no citations have been found about polymerization of PG in the experimental conditions used in this investigation. The exerted capability of PG of retaining enzyme molecules, as well as their activity, has to be probably related to the intermolecular bonds established with the enzyme molecules.

The use of PG as containment net needs to be further investigate, but its capability of preserving LRS, hence the sensitivity of the biosensor over time, is the most important feature in view of a possible prolonged implant for several days, being able to guarantee the most reliable glutamate measurement possible.

At the moment, some experiments are underway to evaluate the role of PG on other biosensor, as glucose and lactate biosensors.”

Round  2

Reviewer 2 Report

The reviewer is pleased with the revised manuscript.